# A Fast Hop-Biased Approximation Algorithm for the Quadratic Group Steiner Tree Problem

## ABSTRACT

Knowledge Graph (KG) exploration helps Web users understand the contents of a large and unfamiliar KG and extract relevant insights. The task has recently been formulated as a Quadratic Group Steiner Tree Problem (QGSTP) to search for a semantically cohesive sub-graph connecting entities that match query keywords. However, on large graphs, existing algorithms for this NP-hard problem cannot meet the performance need. In this paper, we propose a novel approximation algorithm for QGSTP called HB. It finds and merges an optimal set of paths according to a Hop-Biased objective function, which not only leads to a guaranteed approximation ratio but is also decomposable by paths to enable efficient dynamic programming based search. Accompanied by a set of pruning heuristics, HB outperformed the state of the art by 1–2 orders of magnitude, empirically reducing the average time for answering a query on a million-scale graph from about one minute to one second.

## 1 INTRODUCTION

The *Quadratic Group Steiner Tree Problem* (QGSTP) [25, 26], an emerging combinatorial optimization problem on graphs, is a generalization of the famous Group Steiner Tree Problem (GSTP). Given a graph $G$ and a query $\mathbb{Q}$ consisting of $g$ groups (i.e., $g$ subsets of vertices), an optimum answer is a min-cost Quadratic Group Steiner Tree (QGST) which is a sub-tree of $G$ covering all the groups, i.e., containing at least one vertex from each group. The cost of a QGST linearly combines the sum of weights of its vertices—same as the vertex-weighted GSTP [16], and, the sum of *quadratic weights* of its vertex pairs—newly introduced to the cost function by [25].

**Applications.** An important Web application of QGSTP is related to knowledge graph (KG) which represents typed relations as edges between entities as vertices [12]. *KG exploration* [19, 20] is a trending task that helps non-expert Web users comprehend, analyze, and retrieve large and complex KGs, for which keyword query provides a convenient method [24–26]. A user submits a keyword query where each keyword will be matched with a group of vertices as the input of QGSTP. Based on a predefined vertex weighting function characterizing the inverse salience of an entity and a predefined quadratic (i.e., vertex-pair) weighting function characterizing the semantic distance between a pair of entities, a min-cost QGST representing the most salient connection between the query keywords will be found and presented to the user.

Compared with the conventional GSTP, formulating a QGSTP in this scenario helps capture the "semantic cohesiveness" of an answer [4] by *minimizing the sum of quadratic weights representing the pairwise semantic distances between its constituent entities*. For example, Figure 1 illustrates a KG and two QGSTs $T_1$, $T_2$ for a query with two groups. $T_2$ connects the two groups with a set of salient but disparate entities including companies, countries, and a mode of government. It appears semantically disjointed and less meaningful as a whole. By contrast, all the entities in $T_1$ are closely related music

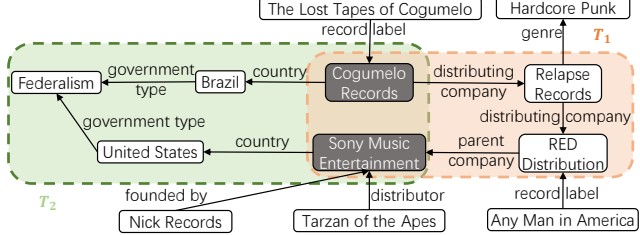

**Figure 1: An example knowledge graph taken from DB-pedia, containing two QGSTs $T_1$ and $T_2$ for the query $\{\{\texttt{Cogumelo Records}\}, \{\texttt{Sony Music Entertainment}\}\}$.**

companies. Such "homogeneous entities" [5] form a semantically cohesive and more meaningful connection which has been proved favourable to users [25]. Solving QGSTP has the potential to find $T_1$, as opposed to $T_2$ found undesirably by solving GSTP.

**Motivation.** Despite its usefulness for Web applications, QGSTP is NP-hard. For this problem, the only exponential-time exact algorithm is $B^3F$ (Branch-and-Bound Best-First) [26], and the polynomial-time approximation algorithm having the currently best approximation ratio is QO (Quality-Oriented) [25]. However, neither of them could scale to graphs containing merely more than tens of thousands of vertices. A more practical algorithm is EO (Efficiency-Oriented) [25]. This polynomial-time approximation algorithm, having a guaranteed approximation ratio of $(g-1)^2 n$ where $n$ is the number of vertices and $g$ is the number of groups, has the capacity to answer a query over a million-scale graph *in about one minute*. Still, such performance cannot meet the needs of KG exploration and other real-time applications that require fast response time [19], e.g., answering a query over a million-scale graph *in one second*.

Our research question is how to design an algorithm that outperforms EO and meets the above performance need and, meantime, has a guaranteed approximation ratio that at least matches EO.

**Our Work.** To solve QGSTP and other GSTP-like problems, a common approximation scheme adopted in the literature [24, 25, 27, 30] is to find and merge a set of paths starting from a common root vertex and ending at vertices from different groups [11, 13], called a *relevant path set* (RPS). Concrete algorithms differ in how to estimate the quality of a RPS, referred to as its hcost in this paper, which is typically designed in such a way that the ratio of the cost of the QGST constructed by merging the paths in a small-hcost RPS to the cost of an optimum QGST is bounded, and, such a small-hcost RPS can be found in polynomial time.

In this paper, we also adopt this scheme but propose a novel *hop-biased* hcost function that prioritizes RPS consisting of small-hop paths, and we prove that it guarantees the same approximation ratio as EO. More importantly, a distinguishing feature of our hcost function is that it is *decomposable by paths*, thus enabling our design of a dynamic programming based path search algorithm called HB

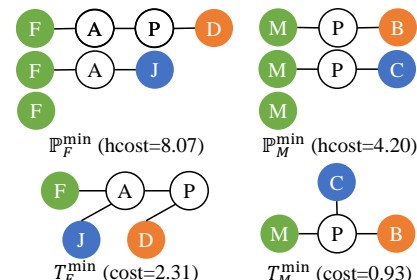

**(a) Graph**

| | w |
|---|---|
| A | 0.40 |
| B | 0.30 |
| C | 0.20 |
| D | 0.30 |
| F | 0.10 |
| J | 0.20 |
| M | 0.10 |
| P | 0.40 |

| qw | A | B | C | D | F | J | M | P |
|---|---|---|---|---|---|---|---|---|
| A | 0 | 0.50 | 0.40 | 0.50 | 0.20 | 0.20 | 0.35 | 0.10 |
| B | 0.50 | 0 | 0.20 | 0.20 | 0.30 | 0.40 | 0.10 | 0.10 |
| C | 0.40 | 0.20 | 0 | 0.25 | 0.50 | 0.45 | 0.10 | 0.20 |
| D | 0.50 | 0.20 | 0.25 | 0 | 0.30 | 0.40 | 0.15 | 0.15 |
| F | 0.20 | 0.30 | 0.50 | 0.30 | 0 | 0.10 | 0.35 | 0.35 |
| J | 0.20 | 0.40 | 0.45 | 0.40 | 0.10 | 0 | 0.40 | 0.40 |
| M | 0.35 | 0.10 | 0.10 | 0.15 | 0.35 | 0.40 | 0 | 0.20 |
| P | 0.10 | 0.10 | 0.20 | 0.15 | 0.35 | 0.40 | 0.20 | 0 |

**(b) Weighting Functions**

**(c) Two RPSes and Two QGSTs**

**Figure 2: Running example with $G$ in (a), w and qw in (b), $\mathbb{Q} = \{\{F, M\}, \{C, J\}, \{B, D\}\}$, $\alpha = 0.3$, and an optimum answer $T_M^{\min}$ in (c).**

(Hop-Biased) for finding a locally min-hcost RPS to be merged. We prove that HB's time complexity significantly improves on EO by a factor of $gn$, and we show that it empirically used less than one second to answer a query over a million-scale graph, thereby meeting the needs of real-time applications including KG exploration. This satisfying performance is also partially attributed to a set of heuristics we develop for pruning the search space.

**Paper Structure.** We formulate QGSTP and RPS in Section 2, detail the HB algorithm in Section 3 and the heuristics in Section 4, and report experimental results in Section 5. We discuss related work in Section 6 before we conclude the paper in Section 7.

## 2 PRELIMINARIES

### 2.1 Problem Formulation

QGSTP is defined on an undirected[1] *graph* $G = \langle V, E \rangle$, where $V$ is a set of vertices and $E \subseteq V \times V$ is a set of edges. For simplicity, we will write $n = |V|$ and $m = |E|$ throughout the paper.

A vertex weighting function $w : V \mapsto \mathbb{R}^{0+}$ assigns each vertex a *weight*, and a quadratic weighting function $qw : V \times V \mapsto \mathbb{R}^{0+}$ assigns each pair of vertices a *quadratic weight*. Following [25], qw is expected to be a pseudometric satisfying that for any $u, v, w \in V$:

$$\mathsf{qw}(v, v) = 0, \ \mathsf{qw}(u, v) = \mathsf{qw}(v, u), \ \mathsf{qw}(u, w) \leq \mathsf{qw}(u, v) + \mathsf{qw}(v, w). \quad (1)$$

Our experiments will use off-the-shelf implementations of w and qw which are orthogonal to our research contribution.

A *query* consists of $g$ groups of vertices $\mathbb{Q} = \{K_1, K_2, \ldots, K_g\}$ in which each $K_i \subseteq V$. A *Quadratic Group Steiner Tree* (QGST) $T = \langle V_T, E_T \rangle$ is a connected and structurally minimal subgraph (i.e., subtree) of $G$ such that $V_T \cap K_i \neq \emptyset$ for each $K_i \in \mathbb{Q}$.

The *cost of a QGST* is defined as a linear combination:

$$\mathsf{cost}(T) = \alpha \cdot \mathsf{cost}_{\mathsf{w}}(T) + \beta \cdot \mathsf{cost}_{\mathsf{qw}}(T), \quad \text{where} \quad (2)$$

$$\mathsf{cost}_{\mathsf{w}}(T) = \sum_{v \in V_T} \mathsf{w}(v),$$

$$\mathsf{cost}_{\mathsf{qw}}(T) = \sum_{\substack{v_i, v_j \in V_T \\ i < j}} \mathsf{qw}(v_i, v_j), \quad (3)$$

and $\alpha$ and $\beta = 1 - \alpha$ are non-negative parameters for controlling the relative importance of weight and quadratic weight.

[1]QGSTP was originally defined on a directed graph but the edges in a path or tree were allowed to be oriented in different directions [25], i.e., essentially undirected.

Given $G$ and $\mathbb{Q}$, solving QGSTP is to find a min-cost QGST as an *optimum answer*. It has been proved that QGSTP is NP-hard [25].

**Running Example.** As illustrated in Figure 2, given the query $\mathbb{Q} = \{\{F, M\}, \{C, J\}, \{B, D\}\}$, the QGST $T_M^{\min}$ with cost = 0.93 is an optimum answer, while $T_F^{\min}$ has a larger cost of 2.31.

### 2.2 Basics: Relevant Path Set (RPS)

The following basic facts about RPS will be used in our algorithm.

**RPS.** A RPS is a set of paths $\mathbb{P}_r = \{p_1, p_2, \ldots, p_g\}$ where each $p_i \in \mathbb{P}_r$ is an $r$-$K_i$ path starting from a given root vertex $r$ and ending at a vertex in a group $K_i$. We refer to it as an $r$-RPS.

**GenAns.** From $\mathbb{P}_r$ we can construct a QGST as follows. Assuming $V_p$ and $E_p$ denote the set of vertices and the set of edges in a path $p$, respectively, we merge the paths in $\mathbb{P}_r$ into a subgraph $G' = \left\langle \bigcup_{p \in \mathbb{P}_r} V_p, \bigcup_{p \in \mathbb{P}_r} E_p \right\rangle$, whose structurally minimal sub-tree is a QGST. We refer to this construction process as $T = \mathsf{GenAns}(\mathbb{P}_r)$.

**GenRPS.** We can also transform a QGST $T = \langle V_T, E_T \rangle$ into an $r$-RPS $\mathbb{P}_r$ for any given root vertex $r \in V_T$. Specifically, for each group $K_i$, since $V_T \cap K_i \neq \emptyset$, we choose from $T$ the path between $r$ and any vertex $v \in V_T \cap K_i$ as an $r$-$K_i$ path $p_i \in \mathbb{P}_r$. We refer to this transformation process as $\mathbb{P}_r = \mathsf{GenRPS}(T, r)$.

**LEMMA 1.** *When $g > 1$, for any QGST $T = \langle V_T, E_T \rangle$, root vertex $r \in V_T$, and RPS $\mathbb{P}_r = \mathsf{GenRPS}(T, r)$, every vertex $v \in V_T \setminus \{r\}$ appears in at most $g - 1$ paths in $\mathbb{P}_r$.*

PROOF. We prove by contradiction. Assume a vertex $v \in V_T \setminus \{r\}$ appears in all the $g$ paths in $\mathbb{P}_r$. For each $r$-$K_i$ path in $\mathbb{P}_r$ between $r$ and $v_i \in K_i$, consider its sub-path between $v$ and $v_i$. All such $g$ sub-paths constitute a $v$-RPS from which we can construct a QGST that is a proper subgraph of $T$, contradicting $T$'s structural minimality. □

**vnum.** Let vnum be the total number of vertices in $\mathbb{P}_r$, where we count the root vertex only once as it appears in every path in $\mathbb{P}_r$:

$$\mathsf{vnum}(\mathbb{P}_r) = 1 + \sum_{p \in \mathbb{P}_r} |V_p \setminus \{r\}|. \quad (4)$$

**Running Example.** In Figure 2, $\mathbb{P}_F^{\min}$ and $\mathbb{P}_M^{\min}$ are two RPSes, each containing three paths. $T_F^{\min}$ and $T_M^{\min}$ are two QGSTs constructed from $\mathbb{P}_F^{\min}$ and $\mathbb{P}_M^{\min}$, respectively, i.e., $T_F^{\min} = \mathsf{GenAns}(\mathbb{P}_F^{\min})$

and $T_M^{\min} = \text{GenAns}(\mathbb{P}_M^{\min})$. We also see $\mathbb{P}_F^{\min} = \text{GenRPS}(T_F^{\min}, F)$, $\mathbb{P}_M^{\min} = \text{GenRPS}(T_M^{\min}, M)$, $\text{vnum}(\mathbb{P}_F^{\min}) = 6$, and $\text{vnum}(\mathbb{P}_M^{\min}) = 5$.

## 3 HOP-BIASED (HB) ALGORITHM

We introduce the main idea of our HB algorithm in Section 3.1, where the key is a new hcost function for estimating the quality of a RPS. We detail the algorithm in Section 3.2, analyze its approximation ratio in Section 3.3, and its time complexity in Section 3.4.

### 3.1 Main Idea

Finding a min-cost QGST requires exponential time. A common approximation scheme for solving such a problem [24, 25, 27, 30] is to search for an optimal RPS in polynomial time and then employ GenAns to merge the paths in this RPS to construct a QGST. The cost of the obtained QGST relies on how an *optimal RPS* is defined. Below we define the hcost of a RPS, which is proved to bound the cost of the QGST constructed from this RPS. This property is important to the proof of approximation ratio of our algorithm in Section 3.2, which efficiently searches for a (locally) min-hcost RPS to be merged into a small-cost QGST.

**hcost of RPS.** According to Eqs. (2)(3), a small-sized QGST is likely to have a small cost. Such a QGST is constructed from a RPS consisting of *small-hop* paths and hence having a small vnum value. Therefore, we include vnum in the definition of hcost of a RPS:

$$\text{hcost}(\mathbb{P}_r) = \alpha \cdot \text{hcost}_w(\mathbb{P}_r) + \beta \cdot \text{hcost}_{qw}(\mathbb{P}_r), \text{ where} \quad (5)$$

$$\text{hcost}_w(\mathbb{P}_r) = \text{vnum}(\mathbb{P}_r)\left(w(r) + \sum_{p \in \mathbb{P}_r} \sum_{v \in V_p \setminus \{r\}} w(v)\right), \quad (6)$$

$$\text{hcost}_{qw}(\mathbb{P}_r) = \text{vnum}(\mathbb{P}_r) \sum_{p \in \mathbb{P}_r} \sum_{v \in V_p \setminus \{r\}} qw(r, v).$$

LEMMA 2. *For any RPS $\mathbb{P}_r$ and the QGST $T = \text{GenAns}(\mathbb{P}_r)$,*

$$\text{cost}(T) \leq \text{hcost}(\mathbb{P}_r). \quad (7)$$

PROOF. We separately prove about w and qw.

$$\text{cost}_w(T) = \sum_{v \in V_T} w(v) \quad \text{[Eq. (3)]}$$

$$\leq w(r) + \sum_{p \in \mathbb{P}_r} \sum_{v \in V_p \setminus \{r\}} w(v) \quad \text{[GenAns]}$$

$$\leq \text{vnum}(\mathbb{P}_r)\left(w(r) + \sum_{p \in \mathbb{P}_r} \sum_{v \in V_p \setminus \{r\}} w(v)\right) \quad (8)$$

$$= \text{hcost}_w(\mathbb{P}_r). \quad \text{[Eq. (6)]}$$

$$\text{cost}_{qw}(T) = \sum_{\substack{v_i, v_j \in V_T \\ i < j}} qw(v_i, v_j) \quad \text{[Eq. (3)]}$$

$$\leq \sum_{\substack{v_i, v_j \in V_T \\ i < j}} qw(v_i, r) + qw(r, v_j) \quad \text{[Eq. (1)]}$$

$$= (|V_T| - 1) \sum_{v \in V_T} qw(r, v) \quad (9)$$

$$\leq \text{vnum}(\mathbb{P}_r) \sum_{p \in \mathbb{P}_r} \sum_{v \in V_p \setminus \{r\}} qw(r, v) \quad \text{[GenAns]}$$

$$= \text{hcost}_{qw}(\mathbb{P}_r). \quad \text{[Eq. (6)]}$$

$$\text{cost}(T) = \alpha \cdot \text{cost}_w(T) + \beta \cdot \text{cost}_{qw}(T) \quad \text{[Eq. (2)]}$$

$$\leq \alpha \cdot \text{hcost}_w(\mathbb{P}_r) + \beta \cdot \text{hcost}_{qw}(\mathbb{P}_r) \quad \text{[Eqs. (8)(9)]} \quad (10)$$

$$= \text{hcost}(\mathbb{P}_r). \quad \text{[Eq. (5)]}$$

$\square$

**Min-hcost RPS.** Since the hcost of a RPS bounds the cost of the QGST constructed from it, we aim at finding a min-hcost RPS. For efficiency considerations, instead of finding a globally min-hcost RPS, our algorithm will search for a locally min-hcost RPS by restricting its root vertex to be within the smallest group $K_{i_{\min}} \in \mathbb{Q}$. More formally, let $\mathbb{P}_r^{\min}$ denote the min-hcost $r$-RPS:

$$\mathbb{P}_r^{\min} = \underset{\mathbb{P}_r}{\arg\min} \, \text{hcost}(\mathbb{P}_r). \quad (11)$$

Our algorithm will construct a QGST from the following RPS:

$$\mathbb{P}^{\#} = \underset{\mathbb{P}_r^{\min}}{\arg\min} \, \text{hcost}(\mathbb{P}_r^{\min}) \quad \text{s.t.} \quad r \in K_{i_{\min}} = \underset{K_i \in \mathbb{Q}}{\arg\min} \, |K_i|. \quad (12)$$

**Running Example.** In Figure 2, assuming $K_{i_{\min}} = \{F, M\}$, for $\text{hcost}(\mathbb{P}_F^{\min}) = 8.07$ and $\text{hcost}(\mathbb{P}_M^{\min}) = 4.20$, Indeed we observe $\text{cost}(T_F^{\min}) \leq \text{hcost}(\mathbb{P}_F^{\min})$ and $\text{cost}(T_M^{\min}) \leq \text{hcost}(\mathbb{P}_M^{\min})$. We have $\mathbb{P}^{\#} = \mathbb{P}_M^{\min}$, and the algorithm will return $T_M^{\min} = \text{GenAns}(\mathbb{P}_M^{\min})$.

### 3.2 Algorithm

Following Eq. (12), the core of our HB algorithm is to search for $\mathbb{P}_r^{\min}$ for each root vertex $r \in K_{i_{\min}}$. For better understanding our bottom-up search process, below we firstly explain its principle in a top-down manner. We only describe the computation of $\text{hcost}(\mathbb{P}_r^{\min})$ since by keeping track of the choices made during the computation we can easily reconstruct $\mathbb{P}_r^{\min}$.

**Principle.** For convenience, we introduce a new function hcost′ to rewrite $\text{hcost}(\mathbb{P}_r)$ defined in Eqs. (5)(6):

$$\text{hcost}(\mathbb{P}_r) = \text{vnum}(\mathbb{P}_r) \cdot \text{hcost}'(\mathbb{P}_r), \text{ where} \quad (13)$$

$$\text{hcost}'(\mathbb{P}_r) = \alpha \cdot w(r) + \sum_{p \in \mathbb{P}_r} \sum_{v \in V_p \setminus \{r\}} \alpha \cdot w(v) + \beta \cdot qw(r, v). \quad (14)$$

With this form, and with $b_{i,j}$ denoting the minimum hcost′ of any *partial* $r$-RPS that covers the first $i$ groups (i.e., $K_1, \ldots, K_i$) with vnum $= j$, we enumerate all possible values of $j$ to compute $\text{hcost}(\mathbb{P}_r^{\min})$:

$$\text{hcost}(\mathbb{P}_r^{\min}) = \min_{1 \leq j \leq g(n-1)+1} j \cdot b_{g,j}, \quad (15)$$

where $j$ is bounded by $g(n - 1) + 1$ because each of the $g$ paths in a RPS has at most $n - 1$ vertices other than the root vertex.

To compute the above $b_{g,j}$, or, more generally, $b_{i,j}$, we decompose hcost′ by paths. Specifically, by slightly abusing the notation, we define the hcost′ of a path:

$$\text{hcost}'(p) = \sum_{v \in V_p \setminus \{r\}} \alpha \cdot w(v) + \beta \cdot qw(r, v), \quad (16)$$

based on which we rewrite $\text{hcost}'(\mathbb{P}_r)$ defined in Eq. (14):

$$\text{hcost}'(\mathbb{P}_r) = \alpha \cdot w(r) + \sum_{p \in \mathbb{P}_r} \text{hcost}'(p). \quad (17)$$

With this form, and with $c_{i,k}$ denoting the minimum hcost′ of any $r$-$K_i$ path that has exactly $k$ edges, we recursively compute $b_{i,j}$:

$$b_{i,j} = \min_{0 \leq k \leq \min\{j-1, \, n-1\}} b_{i-1,j-k} + c_{i,k}, \quad (18)$$

**Algorithm 1:** Hop-Biased (HB) Algorithm

**Input** : Graph $G = \langle V, E \rangle$, Query $\mathbb{Q} = \{K_1, K_2, \ldots, K_g\}$
**Output** : QGST $T^\#$

1   $\mathbb{P}^\# \leftarrow$ null
2   $T^\# \leftarrow$ null
3   $K_{i_{\min}} \leftarrow \underset{K_i \in \mathbb{Q}}{\arg\min} |K_i|$
4   **foreach** $r \in K_{i_{\min}}$ **do**
5     $\mathbb{P}_r^{\min} \leftarrow$ null
6     $d_{0,r} \leftarrow 0$                 // Eq. (20)
7     **foreach** $v \in V \setminus \{r\}$ **do**
8       $d_{0,v} \leftarrow \infty$
9     **for** $k = 1$ **to** $n - 1$ **do**
10       **foreach** $v \in V$ **do**
11         $d_{k,v} \leftarrow \underset{u \in N(v)}{\min} d_{k-1,u} + \alpha \cdot \mathsf{w}(v) + \beta \cdot \mathsf{qw}(r, v)$
12     **for** $i = 1$ **to** $g$ **do**          // Eq. (19)
13       **for** $k = 0$ **to** $n - 1$ **do**
14         $c_{i,k} = \underset{v \in K_i}{\min} d_{k,v}$
15     $b_{0,1} = \alpha \cdot \mathsf{w}(r)$          // Eq. (18)
16     **for** $j = 2$ **to** $g(n - 1) + 1$ **do**
17       $b_{0,j} \leftarrow \infty$
18     **for** $i = 1$ **to** $g$ **do**
19       **for** $j = 1$ **to** $g(n - 1) + 1$ **do**
20         $b_{i,j} \leftarrow \underset{0 \le k \le \min\{j-1,\, n-1\}}{\min} b_{i-1,j-k} + c_{i,k}$
21     **for** $j = 1$ **to** $g(n - 1) + 1$ **do**     // Eq. (15)
22       **if** $\mathbb{P}_r^{\min} =$ null **or** $j \cdot b_{g,j} < \mathsf{hcost}(\mathbb{P}_r^{\min})$ **then**
23         $\mathbb{P}_r^{\min} \leftarrow$ Reconstruct$(b_{g,j})$
24     **if** $\mathbb{P}^\# =$ null **or** $\mathsf{hcost}(\mathbb{P}_r^{\min}) < \mathsf{hcost}(\mathbb{P}^\#)$ **then**
25       $\mathbb{P}^\# \leftarrow \mathbb{P}_r^{\min}$          // Eq. (12)
26     $T_r^{\min} \leftarrow$ GenAns$(\mathbb{P}_r^{\min})$
27     **if** $T^\# =$ null **or** $\mathsf{cost}(T_r^{\min}) < \mathsf{cost}(T^\#)$ **then**
28       $T^\# \leftarrow T_r^{\min}$
29   **return** $T^\#$

where $k$ is bounded by $n - 1$ because a path has at most $n - 1$ edges.

To compute the above $c_{i,k}$, with $d_{k,v}$ denoting the minimum hcost' of any $k$-hop path between $r$ and $v$, we have

$$c_{i,k} = \min_{v \in K_i} d_{k,v}. \tag{19}$$

Finally, we recursively compute $d_{k,v}$:

$$d_{k,v} = \min_{u \in N(v)} d_{k-1,u} + \alpha \cdot \mathsf{w}(v) + \beta \cdot \mathsf{qw}(r, v), \tag{20}$$

where $N(v)$ denotes the set of $v$'s neighbors in the graph.

**Pseudocode.** Algorithm 1 presents the pseudocode of our HB algorithm, which implements the above top-down process of finding $\mathbb{P}^\#$ in a bottom-up manner using dynamic programming. Specifically, for each root vertex $r$ in the smallest group $K_{i_{\min}}$ (lines 3–4), we successively compute all $d_{k,v}$ values according to Eq. (20) (lines 6–11), all $c_{i,k}$ values according to Eq. (19) (lines 12–14), and all $b_{i,j}$ values according to Eq. (18) (lines 15–20), based on which we compute $\mathsf{hcost}(\mathbb{P}_r^{\min})$ according to Eq. (15) and reconstruct $\mathbb{P}_r^{\min}$ (lines 5, 21–23) with the help of some trivial auxiliary arrays for

keeping track of the choices made during the computation. Finally, we obtain a locally min-hcost RPS $\mathbb{P}^\#$ according to Eq. (12) (lines 1, 24–25), and return a correspondingly constructed QGST $T^\#$ (lines 2, 26–29). Here the pseudocode slightly differs from our description in Section 3.1: instead of constructing a single QGST from $\mathbb{P}^\#$, we actually construct and compare $|K_{i_{\min}}|$ QGSTs, one for each root vertex $r \in K_{i_{\min}}$, to achieve a potentially smaller cost (lines 26–28).

**Running Example.** In Figure 2, assuming $K_{i_{\min}} = \{F, M\}$, when $r = F$, the algorithm finds $\mathbb{P}_r^{\min} = \mathbb{P}_F^{\min}$ and constructs $T_r^{\min} = T_F^{\min}$. Then when $r = M$, $\mathbb{P}_r^{\min} = \mathbb{P}_M^{\min}$ is found to have a smaller hcost than $\mathbb{P}_F^{\min}$, and $T_r^{\min} = T_M^{\min}$ has a smaller cost than $T_F^{\min}$, so finally $\mathbb{P}^\# = \mathbb{P}_M^{\min}$, and $T^\# = T_M^{\min}$ is returned.

### 3.3 Approximation Ratio

Theorem 1 shows that our HB algorithm has a guaranteed approximation ratio that matches the state-of-the-art EO algorithm [25].

THEOREM 1. *HB has an approximation ratio of $(g - 1)^2 n$.*

PROOF. Let $T^* = \langle V_{T^*}, E_{T^*} \rangle$ be an optimum answer to QGSTP.

When $g = 1$, $T^*$ contains a single vertex. It is straightforward that $T^\#$ also contains a single vertex and satisfies $\mathsf{cost}(T^\#) = \mathsf{cost}(T^*)$.

When $g > 1$, let $r \in V_{T^*} \cap K_{i_{\min}}$ and $\mathbb{P}_r = \mathsf{GenRPS}(T^*, r)$, then

$$
\begin{aligned}
\mathsf{hcost}_\mathsf{w}(\mathbb{P}_r) &= \mathsf{vnum}(\mathbb{P}_r) \left( \mathsf{w}(r) + \sum_{p \in \mathbb{P}_r} \sum_{v \in V_p \setminus \{r\}} \mathsf{w}(v) \right) && \text{[Eq. (6)]} \\
&\le (g-1)|V_{T^*}| \left( \mathsf{w}(r) + (g-1) \sum_{v \in V_{T^*} \setminus \{r\}} \mathsf{w}(v) \right) && \text{[Lemma 1]} \quad (21) \\
&\le (g-1)^2 |V_{T^*}| \sum_{v \in V_{T^*}} \mathsf{w}(v) \\
&= (g-1)^2 |V_{T^*}| \mathsf{cost}_\mathsf{w}(T^*), && \text{[Eq. (3)]}
\end{aligned}
$$

$$
\begin{aligned}
\mathsf{hcost}_\mathsf{qw}(\mathbb{P}_r) &= \mathsf{vnum}(\mathbb{P}_r) \sum_{p \in \mathbb{P}_r} \sum_{v \in V_p \setminus \{r\}} \mathsf{qw}(r, v) && \text{[Eq. (6)]} \\
&\le (g-1)|V_{T^*}|(g-1) \sum_{v \in V_{T^*}} \mathsf{qw}(r, v) && \text{[Lemma 1]} \\
& && \qquad\qquad (22) \\
&\le (g-1)^2 |V_{T^*}| \sum_{\substack{v_i, v_j \in V_{T^*} \\ i < j}} \mathsf{qw}(v_i, v_j) \\
&= (g-1)^2 |V_{T^*}| \mathsf{cost}_\mathsf{qw}(T^*), && \text{[Eq. (3)]}
\end{aligned}
$$

$$
\begin{aligned}
\mathsf{hcost}(\mathbb{P}_r) &= \alpha \cdot \mathsf{hcost}_\mathsf{w}(\mathbb{P}_r) + \beta \cdot \mathsf{hcost}_\mathsf{qw}(\mathbb{P}_r) && \text{[Eq. (5)]} \\
&\le \alpha \cdot (g-1)^2 |V_{T^*}| \mathsf{cost}_\mathsf{w}(T^*) && \text{[Eq. (21)]} \\
&\quad + \beta \cdot (g-1)^2 |V_{T^*}| \mathsf{cost}_\mathsf{qw}(T^*) && \text{[Eq. (22)]} \quad (23) \\
&\le (g-1)^2 n \cdot (\alpha \cdot \mathsf{cost}_\mathsf{w}(T^*) + \beta \cdot \mathsf{cost}_\mathsf{qw}(T^*)) \\
&= (g-1)^2 n \cdot \mathsf{cost}(T^*). && \text{[Eq. (2)]}
\end{aligned}
$$

This $\mathbb{P}_r$ is in the search space of the algorithm. Therefore,

$$
\begin{aligned}
\mathsf{cost}(T^\#) &\le \mathsf{cost}(\mathsf{GenAns}(\mathbb{P}^\#)) \\
&\le \mathsf{hcost}(\mathbb{P}^\#) && \text{[Lemma 2]} \\
&\le \mathsf{hcost}(\mathbb{P}_r^{\min}) && (24) \\
&\le \mathsf{hcost}(\mathbb{P}_r) \\
&\le (g-1)^2 n \cdot \mathsf{cost}(T^*). && \text{[Eq. (23)]}
\end{aligned}
$$

□

Note that $(g-1)^2 n$ represents the worst case in theory, and $n$ is actually $|V_{T^*}|$. In practical applications of QGSTP such as KG exploration, $|V_{T^*}|$ and $g$ are usually very small. As shown in Section 5.4, the approximation ratio is empirically below 2.18 on real KGs.

## 3.4 Time Complexity

The time complexity of each outermost loop of our HB algorithm (lines 5–28) mainly consists of $O(n(n+m))$ for computing all $d_{k,v}$ values (lines 6–11), $O(gn^2)$ for computing all $c_{i,k}$ values (lines 12–14), $O(g^2 n^2)$ for computing all $b_{i,j}$ values (lines 15–20), $O(g^2 n^2)$ for reconstructing all $\mathbb{P}_r^{\min}$ (lines 21–23), and $O(gn)$ for constructing $T_r^{\min}$ (line 26). The total time complexity of the algorithm is $O(n^2 m + g^2 n^3)$. Compared with the time complexity $O(gn^3 m + g^3 n^4)$ of the state-of-the-art EO algorithm [25], we improve it by a factor of $gn$.

Note that $O(n^2 m + g^2 n^3)$ represents the worst case in theory, and $n$ is actually $|K_{i_{\min}}|$ or caps the number of hops of an $r$-$K_i$ path in a RPS. In practical applications of QGSTP such as KG exploration, these numbers as well as $g$ are usually very small. By further incorporating the heuristics in Section 4 for pruning the search space, as shown in Section 5.2, the average runtime is empirically below 0.84 second on real KGs containing millions of vertices and edges.

## 4 PRUNING HEURISTICS

In this section, we present three heuristics for pruning the search space of our HB algorithm. Despite not changing the worst-case time complexity of the algorithm, as shown in Section 5.5, empirically they considerably improve the efficiency of the algorithm.

## 4.1 Pruning Roots (PR)

In Algorithm 1, each vertex $r \in K_{i_{\min}}$ is considered as the root vertex of a RPS (line 4). Our first pruning heuristic is to, at the beginning of each outermost loop of the algorithm, inexpensively compute a *lower bound* $\mathcal{L}(r)$ for $\text{hcost}(\mathbb{P}_r^{\min})$ and, if $\mathcal{L}(r) \geq \text{hcost}(\mathbb{P}^\#)$, we will safely skip the current loop since it will not update $\mathbb{P}^\#$ (line 24).

To compute $\mathcal{L}(r)$, we calculate the minimum vnum of an $r$-RPS:

$$h_r = 1 + \sum_{i=1}^{g} h_{r,i}, \tag{25}$$

where $h_{r,i}$ denotes the smallest number of hops between $r$ and $K_i$, which is efficiently computed by looking up a *hub labeling* index [1]. The look-up time is in $O(n^2)$ and is practically close to $\Theta(|K_i|)$, or even shorter if using a dynamic version of the index [24]. With $h_r$, we introduce two implementations of $\mathcal{L}(r)$.

Our first implementation of $\mathcal{L}(r)$, denoted by $\mathcal{L}^{(1)}(r)$, is

$$\mathcal{L}^{(1)}(r) = h_r \left( \alpha \cdot w(r) + \sum_{i=1}^{g} \mathcal{L}_i^{(1)}(r) \right), \text{ where}$$
$$\mathcal{L}_i^{(1)}(r) = \min_{p \in \mathcal{P}_{r,i}} \text{hcost}'(p), \tag{26}$$

in which $\mathcal{P}_{r,i}$ denotes the set of all $r$-$K_i$ paths. Comparing this equation with Eqs. (13)(17), we verify that $\mathcal{L}^{(1)}(r) \leq \text{hcost}(\mathbb{P}_r^{\min})$.

We compute $\mathcal{L}_i^{(1)}(r)$ by exploiting the definition of $\text{hcost}'(p)$ in Eq. (16) which resembles the length of a path in an edge-weighted graph. Accordingly, $\mathcal{L}_i^{(1)}(r)$ resembles the distance between $r$ and $K_i$, and is computed by Dijkstra's algorithm in $O(m + n \log n)$

time. We will prune the search space of Dijkstra's algorithm in Section 4.2.

Our second implementation of $\mathcal{L}(r)$, denoted by $\mathcal{L}^{(2)}(r)$, is

$$\mathcal{L}^{(2)}(r) = h_r \left( \alpha \cdot w(r) + \sum_{i=1}^{g} \mathcal{L}_i^{(2)}(r) \right), \text{ where}$$
$$\mathcal{L}_i^{(2)}(r) = \min_{p \in \mathcal{P}_{r,i}} \alpha \cdot \sum_{v \in V_p \setminus \{r\}} w(v) + \beta \cdot \frac{1}{2} \sum_{(u,v) \in E_p} \text{qw}(u,v). \tag{27}$$

According to the triangle inequality given in Eq. (1),

$$\frac{1}{2} \sum_{(u,v) \in E_p} \text{qw}(u,v) \leq \frac{1}{2} \sum_{(u,v) \in E_p} \text{qw}(u,r) + \text{qw}(r,v) \leq \sum_{v \in V_p \setminus \{r\}} \text{qw}(r,v), \tag{28}$$

so with the definition of $\text{hcost}'(p)$ in Eq. (16), we verify that

$$\mathcal{L}_i^{(2)}(r) \leq \min_{p \in \mathcal{P}_{r,i}} \text{hcost}'(p) = \mathcal{L}_i^{(1)}(r), \tag{29}$$

which indicates that $\mathcal{L}^{(2)}(r)$ is a looser lower bound than $\mathcal{L}^{(1)}(r)$.

We compute $\mathcal{L}_i^{(2)}(r)$ by exploiting its definition in Eq. (27) which resembles the distance between $r$ and $K_i$ in a graph with both vertex weights and edge weights, which again is efficiently computed by looking up a (dynamic) hub labeling index in $O(n^2)$ time.

We add $\mathcal{L}^{(1)}(r)$ and $\mathcal{L}^{(2)}(r)$ to Algorithm 1 as follows. At the beginning of each outermost loop (line 4), we compute $\mathcal{L}^{(2)}(r)$ as it is computationally less expensive than $\mathcal{L}^{(1)}(r)$. If $\mathcal{L}^{(2)}(r) \geq \text{hcost}(\mathbb{P}^\#)$, we will skip the current loop, or else we compute $\mathcal{L}^{(1)}(r)$, and will skip the current loop if $\mathcal{L}^{(1)}(r) \geq \text{hcost}(\mathbb{P}^\#)$.

## 4.2 Pruning Paths (PP)

In Section 4.1, the computation of $\mathcal{L}^{(1)}$ relies on Dijkstra's algorithm. Our second pruning heuristic is to maintain, during the execution of Dijkstra's algorithm, an *upper bound* $\mathcal{U}(r)$ for all uncomputed $\mathcal{L}_i^{(1)}(r)$. If exceeded, it would ensure $\mathcal{L}_i^{(1)}(r) \geq \text{hcost}(\mathbb{P}^\#)$ so that we could prune the remaining search space of Dijkstra's algorithm and safely skip the current outermost loop of Algorithm 1.

Specifically, following the properties of Dijkstra's algorithm, $\mathcal{L}_i^{(1)}(r)$ is computed when visiting any vertex in $K_i$ for the first time. With $U = \{1, 2, \ldots, g\}$ and $S$ denoting all $i \in U$ such that $\mathcal{L}_i^{(1)}(r)$ has been computed, based on the definition of $\mathcal{L}^{(1)}(r)$ in Eq. (26), $\mathcal{L}_i^{(1)}(r) < \text{hcost}(\mathbb{P}^\#)$ would imply that for each $i \in U \setminus S$:

$$\mathcal{L}_i^{(1)}(r) < \frac{\text{hcost}(\mathbb{P}^\#)}{h_r} - \alpha \cdot w(r) - \sum_{j \in S} \mathcal{L}_j^{(1)}(r) - \sum_{j \in U \setminus (S \cup \{i\})} \mathcal{L}_j^{(1)}(r)$$
$$\leq \frac{\text{hcost}(\mathbb{P}^\#)}{h_r} - \alpha \cdot w(r) - \sum_{j \in S} \mathcal{L}_j^{(1)}(r) - \sum_{j \in U \setminus (S \cup \{i\})} \mathcal{L}_j^{(2)}(r)$$
$$\leq \frac{\text{hcost}(\mathbb{P}^\#)}{h_r} - \alpha \cdot w(r) - \sum_{j \in S} \mathcal{L}_j^{(1)}(r) - \left( \sum_{j \in U \setminus S} \mathcal{L}_j^{(2)}(r) - \max_{j \in U \setminus S} \mathcal{L}_j^{(2)}(r) \right). \tag{30}$$

The last line is independent of $i$ and we let it be $\mathcal{U}(r)$. We maintain $\mathcal{U}(r)$ during the execution of Dijkstra's algorithm and, if the $\text{hcost}'$ of the current path exceeds $\mathcal{U}(r)$, following the monotonicity of Dijkstra's algorithm, every uncomputed $\mathcal{L}_i^{(1)}(r)$ (i.e., $i \in U \setminus S$) will also exceed $\mathcal{U}(r)$, thus preventing $\mathcal{L}_i^{(1)}(r) < \text{hcost}(\mathbb{P}^\#)$.

We compute $\mathcal{U}(r)$ in $O(g)$ time, given that $h_r$, $\mathcal{L}_j^{(1)}(r)$, and $\mathcal{L}_j^{(2)}(r)$ have been computed in Section 4.1.

**Table 1: Graphs and Queries**

| Graph | $n$ | $m$ | #queries | $g$ |
|---|---|---|---|---|
| MND | 43,750 | 128,402 | 39 | 1−4 |
| CYC | 120,657 | 425,643 | 50 | 1−6 |
| LMDB | 1,326,987 | 2,873,140 | 200 | 1−10 |
| YAGO | 2,215,203 | 5,428,229 | 32 | 2−6 |
| DBP | 5,707,071 | 18,591,110 | 330 | 2−6 |
| LUBM-10U | 207,631 | 811,556 | 400 | 2−16 |
| LUBM-50U | 1,083,833 | 4,243,017 | 400 | 2−16 |
| LUBM-250U | 5,426,913 | 22,281,357 | 400 | 2−16 |
| LUBM-2U | 38,384 | 154,039 | 50 | 4 |
| DBP-50K | 50,000 | 180,369 | 183 | 2−6 |

## 4.3 Principled Initialization (PI)

The pruning heuristics presented in Section 4.1 and Section 4.2 both rely on the current $\mathbb{P}^\#$. Our third pruning heuristic is to inexpensively *initialize* $\mathbb{P}^\#$ to have a small hcost to improve the possibility of applying the above two pruning heuristics.

Specifically, for each root vertex $r \in K_{i_{\min}}$, we find a min-vnum RPS (i.e., vnum = $h_r$) where each $h_{r,i}$-hop $r$-$K_i$ path has the smallest hcost′. Recall that the definition of hcost′($p$) in Eq. (16) resembles the length of a path in an edge-weighted graph. Therefore, the above $g$ paths in a RPS are found by performing BFS-like search in $O(n+m+g)$ time. Among these $|K_{i_{\min}}|$ RPSes, we choose the one having the smallest hcost to initialize $\mathbb{P}^\#$ and use its correspondingly constructed QGST to initialize $T^\#$ in Algorithm 1 (lines 1–2).

## 5 EXPERIMENTS

We evaluated the efficiency, scalability, and effectiveness of our HB algorithm. Following the experimental design in the literature [25, 26], we evaluated in the context of applying QGSTP to keyword-based KG exploration. All the algorithms in the experiments were implemented in Java and executed on an Intel Xeon E5-2643 v4 CPU (3.40GHz) with 180GB memory available for Java programs.

Our code and data are available on Anonymous GitHub.[2]

### 5.1 Experimental Setup

The following setup was used in all the experiments.

**Weighting Functions w and qw.** Following [25], we weighted each vertex by inverting its normalized PageRank score, and we defined the quadratic weight of each pair of vertices (i.e., entities) as the angular distance between their ten-dimensional KG embedding vectors precomputed by RDF2Vec [23].

**Settings of $\alpha$.** Following [25], we used $\alpha \in \{0.1, 0.5, 0.9\}$.

**Baselines.** We compared with EO [25], a state-of-the-art approximation algorithm for QGSTP. We did not compare with B$^3$F [26] or QO [25] because they were only of theoretical interest and could not scale to more than tens of thousands of vertices in practice.

### 5.2 Experiment 1: Efficiency

This experiment evaluated the runtime of HB in real settings.

---

[2]https://anonymous.4open.science/r/QGSTP-HB-EFCC

**Table 2: Mean Runtime per Query (in Seconds) and Proportion of Timeout Queries**

| Graph | Algorithm | $\alpha = 0.1$ | $\alpha = 0.5$ | $\alpha = 0.9$ |
|---|---|---|---|---|
| MND | HB | 0.02 (0.00%) | 0.01 (0.00%) | 0.02 (0.00%) |
| | EO | 0.29 (0.00%) | 0.26 (0.00%) | 0.36 (0.00%) |
| CYC | HB | 0.10 (0.00%) | 0.09 (0.00%) | 0.07 (0.00%) |
| | EO | 3.19 (0.00%) | 3.38 (0.00%) | 4.12 (0.00%) |
| LMDB | HB | 0.13 (0.00%) | 0.08 (0.00%) | 0.07 (0.00%) |
| | EO | 8.83 (0.00%) | 7.37 (0.00%) | 9.81 (0.00%) |
| YAGO | HB | 0.16 (0.00%) | 0.12 (0.00%) | 0.41 (0.00%) |
| | EO | 12.46 (0.00%) | 18.06 (0.00%) | 55.81 (3.13%) |
| DBP | HB | 0.84 (0.00%) | 0.54 (0.00%) | 0.65 (0.00%) |
| | EO | 50.94 (3.33%) | 57.73 (5.45%) | 113.74 (37.88%) |

**Graphs.** We used all the real KGs used in the literature [25, 26]: two small-scale graphs Mondial[3] (MND) and OpenCyc[4] (CYC), two medium-scale graphs LinkedMDB[5] (LMDB) and YAGO[6], and a large-scale graph DBpedia[7] (DBP). Their sizes are shown in Table 1.

**Queries.** Firstly we followed [25] to collect keyword queries. Specifically, for MND, we reused the keyword queries from [6]. For CYC, we generated 50 keyword queries containing 1–6 keywords randomly sampled from the KG. For LMDB, we randomly sampled 200 natural language questions from [22] and removed stop words and punctuation marks. For YAGO, we reused the keyword queries from [29]. For DBP, we reused the keyword queries from [9] and excluded those containing a single keyword.

Then we converted each keyword query to a query in the formulation of QGSTP by mapping each keyword to a group of vertices (i.e., entities) whose name (i.e., rdfs:label) contained that keyword. Queries containing empty groups were removed. Some statistics about the remaining queries are shown in Table 1.

**Metrics.** We ran each algorithm for each query on each graph and reported its runtime. To avoid unacceptably long runtime, we would cancel a run when 200 seconds had passed without completion. Such timeout runs were excluded when reporting mean runtime, which actually benefited the reported runtime of the baseline algorithm while our HB algorithm never encountered timeout.

**Results.** Table 2 shows, for each algorithm, the mean runtime for a query it used and the proportion of timeout queries it encountered. On small-scale graphs MND and CYC, our HB algorithm answered a query within 0.1 second on average, while the EO algorithm used more than 3 seconds on CYC. On medium-scale graphs LMDB and YAGO, while EO spent 7.37−55.81 seconds per query, HB still responded within only 0.41 second. On the large-scale graph DBP, EO suffered timeout on 3.33−37.88% of the queries, and used an average of more than 50 seconds to answer the remaining queries; by contrast, HB never encountered timeout and its mean runtime was at most 0.84 second. To conclude, our HB algorithm used an

---

[3]https://www.dbis.informatik.uni-goettingen.de/Mondial/Mondial-RDF/mondial.rdf
[4]https://master.dl.sourceforge.net/project/texai/open-cyc-rdf/1.1/open-cyc.rdf.ZIP?viasf=1
[5]https://www.cs.toronto.edu/~oktie/linkedmdb/linkedmdb-latest-dump.zip
[6]https://yago-knowledge.org/data/yago1/yago-1.0.0-turtle.7z
[7]https://downloads.dbpedia.org/2016-10/core/mappingbased_objects_en.ttl.bz2

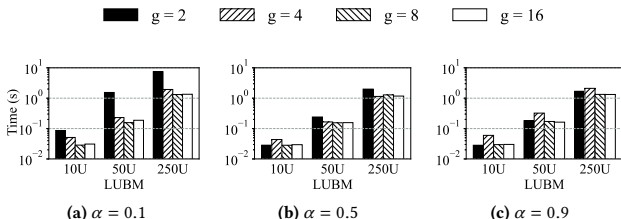

**Figure 3: Mean runtime per query of HB w.r.t. varying $g$.**

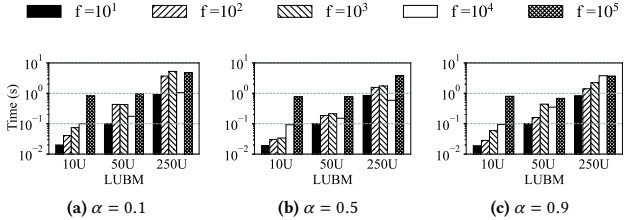

**Figure 4: Mean runtime per query of HB w.r.t. varying $f$.**

average of less than one second to answer a query over a million-scale graph, and was empirically 1–2 orders of magnitude (15–175x) as fast as the state-of-the-art EO algorithm. *These results demonstrated the efficiency of HB and its promising performance to meet the needs of real-time applications.*

Appendix A presents the distribution of runtime over all queries. Our HB algorithm very rarely used more than one second for a query, demonstrating its robust performance.

### 5.3 Experiment 2: Scalability

This experiment evaluated the runtime of HB w.r.t. varying input sizes. According to the analysis in Section 3.4, the runtime of HB is related to the order and size of a graph (i.e., $n$ and $m$), the number of groups in a query (i.e., $g$), and the size of a group (e.g., $|K_{i_{\min}}|$).

**Graphs.** We followed [25] to use LUBM's KG generator[8] to generate three graphs of linearly increasing sizes: LUBM-10U, LUBM-50U, and LUBM-250U. Their sizes are shown in Table 1.

**Queries.** We followed [25] to generate queries by exponentially varying two parameters: the number of groups $g$ and the size of each group $f$. Specifically, for each combination $(g, f) \in \{(2, 10^2), (4, 10^2), (8, 10^2), (16, 10^2)\}$ and each $(g, f) \in \{(4, 10^1), (4, 10^2), (4, 10^3), (4, 10^4), (4, 10^5)\}$, we generated 50 queries consisting of $g$ groups of $f$ vertices randomly sampled from the graph.

**Metrics.** We ran our HB algorithm for each query on each graph and reported its runtime.

**Results.** Figure 3 shows the mean runtime used by HB for a query under different $g$. Possibly contrary to intuition, by increasing $g$, the runtime did not rise but drop in many settings. We attributed it to our pruning heuristic PI described in Section 4.3. Specifically, when $g$ was larger, the locally min-hcost RPS $\mathbb{P}^\#$ that HB searched for was more likely to consist of min-hop paths between groups, which were exactly those found by PI to initialize $\mathbb{P}^\#$, so the search space of the HB algorithm would be pruned faster.

[8]http://swat.cse.lehigh.edu/projects/lubm/

**Table 3: Mean Empirical Approximation Ratio per Query**

| Graph | Algorithm | $\alpha = 0.1$ | $\alpha = 0.5$ | $\alpha = 0.9$ |
|---|---|---|---|---|
| LUBM-2U | HB | $1.13 \pm 0.16$ | $1.12 \pm 0.17$ | $1.06 \pm 0.11$ |
| | EO | $1.20 \pm 0.26$ | $1.15 \pm 0.18$ | $1.08 \pm 0.12$ |
| DBP-50K | HB | $1.07 \pm 0.15$ | $1.07 \pm 0.14$ | $1.07 \pm 0.13$ |
| | EO | $1.09 \pm 0.19$ | $1.09 \pm 0.18$ | $1.04 \pm 0.08$ |

Figure 4 shows the mean runtime used by HB for a query under different $f$. It was not surprising that the runtime grew by increasing $f$ since the outermost loop of the HB algorithm was executed $f$ times. Thanks to pruning, the growth appeared sublinear, e.g., by increasing $f$ from $10^1$ to $10^5$ by 4 orders of magnitude, the runtime only grew by about 1–2 orders of magnitude.

In Figure 3 and Figure 4, by increasing $n$ and $m$ from LUBM-10U to 250U by 25x, the runtime grew by 5–91x with an average of 42x, representing a roughly linear increase. It was much better than the worst-case cubic time complexity in theory proved in Section 3.4. *These results demonstrated the scalability of HB w.r.t. input size.*

### 5.4 Experiment 3: Effectiveness

This experiment evaluated the practical approximation ratio of HB.

**Graphs.** Measuring approximation ratio required knowing the cost of an optimum answer which could be found by an exponential-time exact algorithm for QGSTP such as $B^3F$ [26]. However, due to its limited scalability, we had to follow [25] to use two small KGs for this experiment: LUBM-2U generated by LUBM, and a subgraph DBP-50K extracted from DBP. Their sizes are shown in Table 1.

**Queries.** For LUBM-2U, we generated queries in the same way as we did in Section 5.3 under $g = 4$ and $f = 10^2$. For DBP-50K, we obtained queries in the same way as we did in Section 5.2 for DBP.

**Metrics.** We ran each approximation algorithm for each query on each graph and reported its empirical approximation ratio:

$$\text{empirical approximation ratio} = \frac{\text{cost}(T^\#)}{\text{cost}(T^*)}, \tag{31}$$

where $\text{cost}(T^\#)$ and $\text{cost}(T^*)$ represented the QGSTs found by the approximation algorithm and the exact $B^3F$ algorithm, respectively.

**Results.** Table 3 shows, for each approximation algorithm, its mean empirical approximation ratio for a query with standard deviation. Our HB algorithm and the EO algorithm achieved similar and satisfying approximation ratios. HB achieved a mean empirical approximation ratio of at most 1.13. Its largest ratio on a single query was only 2.18. It was much better than the worst-case cubic approximation ratio in theory proved in Section 3.3. *These results demonstrated the effectiveness of HB which matched that of EO.*

### 5.5 Experiment 4: Ablation Study

This experiment evaluated the usefulness of our heuristics for pruning the search space of HB.

**Graphs, Queries, and Metrics.** We reused the settings in Section 5.2 and reported the runtime of each algorithm for each query on each graph. Due to space limitations, we reported the results on LMDB below and the results on other graphs in Appendix B. The conclusions derived from different graphs were consistent.

**Table 4: Mean Runtime per Query (in Seconds) and Proportion of Timeout Queries on LMDB**

| Algorithm | $\alpha = 0.1$ | $\alpha = 0.5$ | $\alpha = 0.9$ |
|---|---|---|---|
| HB | 0.13 (0.00%) | 0.08 (0.00%) | 0.07 (0.00%) |
| HB w/o pruning | 17.46 (0.01%) | 17.18 (0.01%) | 18.10 (0.01%) |
| EO w/o pruning | - (100.00%) | - (100.00%) | - (100.00%) |
| HB w/o PP | 0.19 (0.00%) | 0.10 (0.00%) | 0.09 (0.00%) |
| HB w/o PI | 0.57 (0.00%) | 0.42 (0.00%) | 0.47 (0.00%) |

**Results.** Table 4 shows, for each algorithm, the mean runtime for a query it used and the proportion of timeout queries it encountered on LMDB. Without pruning, the runtime of our HB algorithm rose by 2 orders of magnitude, but still, it was much faster than the corresponding pruning-free version of the EO algorithm which suffered timeout on all the queries. By disabling a single pruning heuristic PP or PI,[9] the runtime of HB also rose considerably by at least 25% and 338%, respectively. *These results demonstrated the usefulness of our pruning heuristics, and also revealed that HB's performance advantage over EO was primarily sourced from our algorithmic advancement rather than from our pruning heuristics.*

## 6 RELATED WORK

### 6.1 Quadratic Group Steiner Tree Problem

The extension from GSTP to QGSTP brings advantages as well as new research challenges. Table 5 compares known algorithms for QGSTP. B³F [26] is the only exact algorithm for computing an optimum answer. Despite using a branch-and-bound strategy, due to the NP hardness of QGSTP, it requires exponential time and only works on small-scale graphs. QO [25] is an approximation algorithm with the currently best approximation ratio but cannot scale to large graphs. EO [25] sacrifices its approximation ratio to reduce its search space and can work on large-scale graphs. However, it needs about one minute to answer a query on a million-scale graph, which is insufficient for real-time applications.

While our HB algorithm and the state-of-the-art EO algorithm both find and merge a RPS, they differ in how the quality of a RPS is estimated and accordingly in the search procedure. Compared with pcost—the cost function for RPS used with EO (and QO), our novel hcost more skillfully incorporates $\mathsf{vnum}(\mathbb{P}_r)$ in such a way that by decomposing the hcost of a RPS into the hcost' of a set of paths via Eqs. (13)(17), the latter is independent of $\mathsf{vnum}(\mathbb{P}_r)$ as shown in Eq. (16). By contrast, in EO, $\mathsf{vnum}(\mathbb{P}_r)$ is tightly coupled with pcost. As a result, for each root vertex $r \in K_{i_{\min}}$, our HB algorithm computes each $d_{k,v}$ and $b_{i,j}$ only once, whereas their counterparts in EO have to be computed $\Theta(gn)$ times, each time with a distinct possible value of $\mathsf{vnum}(\mathbb{P}_r)$. This key improvement leads to HB's significant performance advantage over EO while their approximation ratios are similar.

---

[9]The pruning heuristic PR cannot be disabled alone since PP and PI rely on it.

**Table 5: Known Algorithms for QGSTP**

| Algorithm | Approximation Ratio | Time Complexity |
|---|---|---|
| B³F [26] | 1 | Exponential time |
| QO [25] | $(g-1)^2$ | $O(gn^3m + g^3n^4)$ |
| EO [25] | $(g-1)^2 n$ | $O(gn^3m + g^3n^4)$ |
| Our HB | $(g-1)^2 n$ | $O(n^2m + g^2n^3)$ |

### 6.2 Group Steiner Tree Problem

QGSTP generalizes from GSTP, an established combinatorial optimization problem on graphs having $O(\log^{2-\epsilon} g)$ inapproximability [8]. Exact algorithms include dynamic programming [7, 18]. Scalable algorithms [3, 24, 27] usually adopt the One-Star approximation scheme [11, 13] to find and merge a RPS, and have a guaranteed approximation ratio of $g - 1$. Algorithms having a better approximation ratio [2] cannot scale to large graphs.

It would be meaningless to directly apply these algorithms to QGSTP since they could not handle quadratic weights in Eq. (2) and hence may return arbitrarily bad results, e.g., when $\alpha = 0$.

### 6.3 Keyword-Based KG Exploration

QGSTP was originally proposed to formulate keyword-based KG exploration [25, 26]. Prior to that, the task was formulated as GSTP [3, 21, 24, 29] or its variants [10, 14, 15, 17, 28]. QGSTP incorporates quadratic weights to represent semantic distances between entities in a KG, which helps find semantically cohesive answers.

Another recent extension of GSTP bounds the diameter of an answer [30], which is orthogonal to our work on QGSTP.

## 7 CONCLUSION

QGSTP has showed in the literature its usefulness in formulating keyword-based KG exploration, but was lacking efficient solutions. Our work fills the gap with the proposed HB algorithm. Building on a novel hop-biased hcost function for RPS which is decomposable by paths and hence allows for a fast dynamic programming based search procedure, HB outperforms the state-of-the-art EO algorithm by a factor of $gn$ in theory and by 1–2 orders of magnitude in the experiments. With an approximation ratio comparable with EO, HB only used an average of less than one second to answer a query over a million-scale graph, meeting the performance need of real-time applications. It will enable the Web community to expand the range of potential applications of QGSTP as a generalization of the popularly used GSTP to offer increased expressiveness and utility.

In future work, we will consider three directions. From the algorithm perspective, whereas HB has exhibited a satisfying empirical approximation ratio in the experiments, we plan to cautiously extend its search space to seek a better approximation ratio guarantee while maintaining its current efficiency and scalability. It will help further widen the application of QGSTP. From the problem perspective, we will analyze the inapproximability of QGSTP to explore whether it is fundamentally harder than GSTP. It will give us more insights into this emerging NP-hard problem. From the application perspective, beyond KG exploration, we will investigate existing Web applications of GSTP to identify opportunities for enhancement with QGSTP and our HB algorithm.

## A EXTENDED RESULTS OF EXPERIMENT 1: EFFICIENCY

Figure 5 shows, for each algorithm, the cumulative distribution of its runtime over all queries. For a considerable proportion of the queries, the EO algorithm responded after more than one second on small-scale graphs MND and CYC, more than ten seconds on medium-scale graphs LMDB and YAGO, and more than one hundred seconds on the large-scale graph DBP. By contrast, our HB algorithm very rarely used more than one second for a query. Even on DBP, it spent two or more seconds only on 11%, 4%, and 11% of the queries under $\alpha = 0.1$, $\alpha = 0.5$, and $\alpha = 0.9$, respectively. *These results demonstrated the robust performance of HB.*

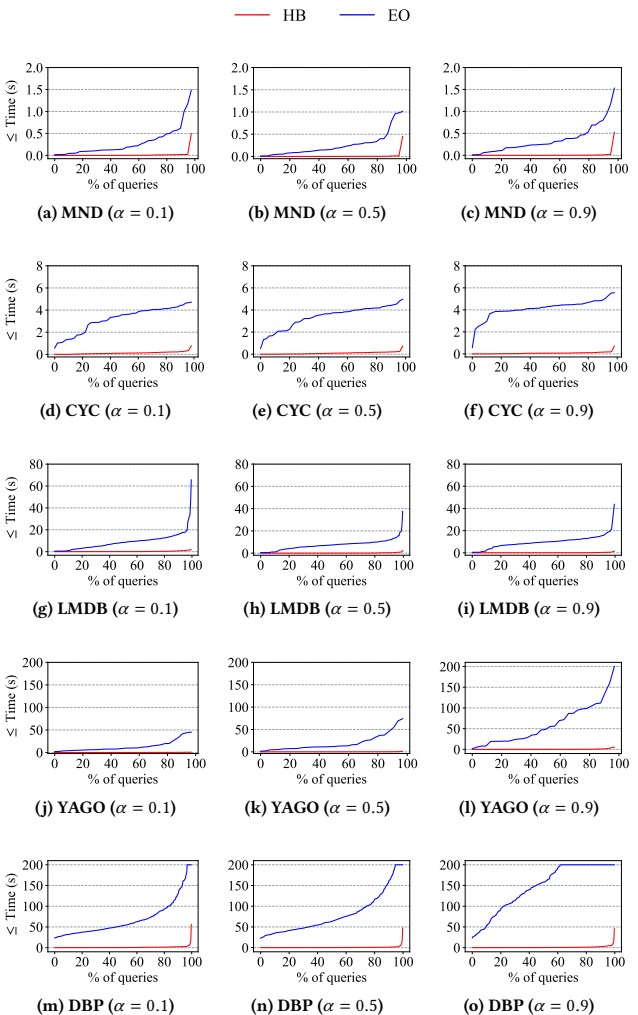

**(a) MND ($\alpha = 0.1$)**   **(b) MND ($\alpha = 0.5$)**   **(c) MND ($\alpha = 0.9$)**

**(d) CYC ($\alpha = 0.1$)**   **(e) CYC ($\alpha = 0.5$)**   **(f) CYC ($\alpha = 0.9$)**

**(g) LMDB ($\alpha = 0.1$)**   **(h) LMDB ($\alpha = 0.5$)**   **(i) LMDB ($\alpha = 0.9$)**

**(j) YAGO ($\alpha = 0.1$)**   **(k) YAGO ($\alpha = 0.5$)**   **(l) YAGO ($\alpha = 0.9$)**

**(m) DBP ($\alpha = 0.1$)**   **(n) DBP ($\alpha = 0.5$)**   **(o) DBP ($\alpha = 0.9$)**

**Figure 5: Cumulative distribution of runtime over all queries.**

## B EXTENDED RESULTS OF EXPERIMENT 4: ABLATION STUDY

Table 6 shows, for each algorithm, the mean runtime for a query it used and the proportion of timeout queries it encountered on small-scale and medium-scale graphs. Without pruning, the runtime of our HB algorithm rose on all the three graphs, but still, it was much faster than the corresponding pruning-free version of the EO algorithm which suffered timeout on all the queries. On larger graphs like YAGO and DBP, HB without pruning also suffered timeout on most queries.

Table 7 shows, for each algorithm, the mean runtime for a query it used and the proportion of timeout queries it encountered on medium-scale and large-scale graphs where the search space was relatively large and the effects of pruning could be easily observed. By disabling a single pruning heuristic PP or PI, the runtime of HB rose considerably by at least 13% and 49%, respectively.

These results were consistent with the conclusions in Section 5.5.

**Table 6: Mean Runtime per Query (in Seconds) and Proportion of Timeout Queries on MND, CYC, and LMDB**

| Graph | Algorithm | $\alpha = 0.1$ | $\alpha = 0.5$ | $\alpha = 0.9$ |
|---|---|---|---|---|
| MND | HB | 0.02 (0.00%) | 0.01 (0.00%) | 0.02 (0.00%) |
| | HB w/o pruning | 0.07 (0.00%) | 0.06 (0.00%) | 0.07 (0.00%) |
| | EO w/o pruning | - (100.00%) | - (100.00%) | - (100.00%) |
| CYC | HB | 0.10 (0.00%) | 0.09 (0.00%) | 0.07 (0.00%) |
| | HB w/o pruning | 0.19 (0.00%) | 0.19 (0.00%) | 0.20 (0.00%) |
| | EO w/o pruning | - (100.00%) | - (100.00%) | - (100.00%) |
| LMDB | HB | 0.13 (0.00%) | 0.08 (0.00%) | 0.07 (0.00%) |
| | HB w/o pruning | 17.46 (0.01%) | 17.18 (0.01%) | 18.10 (0.01%) |
| | EO w/o pruning | - (100.00%) | - (100.00%) | - (100.00%) |

**Table 7: Mean Runtime per Query (in Seconds) and Proportion of Timeout Queries on LMDB, YAGO, and DBP**

| Graph | Algorithm | $\alpha = 0.1$ | $\alpha = 0.5$ | $\alpha = 0.9$ |
|---|---|---|---|---|
| LMDB | HB | 0.13 (0.00%) | 0.08 (0.00%) | 0.07 (0.00%) |
| | HB w/o PP | 0.19 (0.00%) | 0.10 (0.00%) | 0.09 (0.00%) |
| | HB w/o PI | 0.57 (0.00%) | 0.42 (0.00%) | 0.47 (0.00%) |
| YAGO | HB | 0.16 (0.00%) | 0.12 (0.00%) | 0.41 (0.00%) |
| | HB w/o PP | 0.18 (0.00%) | 0.15 (0.00%) | 1.22 (0.00%) |
| | HB w/o PI | 0.24 (0.00%) | 0.33 (0.00%) | 4.61 (0.00%) |
| DBP | HB | 0.84 (0.00%) | 0.54 (0.00%) | 0.65 (0.00%) |
| | HB w/o PP | 1.27 (0.00%) | 0.94 (0.00%) | 1.55 (0.00%) |
| | HB w/o PI | 1.25 (0.00%) | 1.16 (0.00%) | 5.52 (0.00%) |

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
