# OpenReview forum: "A Fast Hop-Biased Approximation Algorithm for the Quadratic Group Steiner Tree Problem"
_ACM.org/TheWebConf/2024/Conference — TheWebConf24 Oral_

### Official Review · Reviewer_7mV9 · 2023-11-17

**Novelty:** 4
**Technical Quality:** 4

**Review:**

Summary of contribution

In this paper, the authors target on the Quadratic Group Steiner Tree Problem (QGSTP), aiming to search for a semantically cohesive sub-graph connecting entities that match query keywords. They propose an approximation algorithm HB, which finds and merges paths based on hop-biased objective function. It is empirically efficient comparing with existing methods on several datasets.

Strong points

S1. This paper’s main goal is clear, i.e., designing an algorithm that outperforms existing methods in efficiency and achieves the similar or same approximation ratio with them.

S2. Technique parts are clear and self-contained.

S3. Extensive experiments are conducted to prove the proposed techniques’ effectiveness.

Weak points

W1. The motivation is weak. QGST is a highly abstract problem that may be important in some areas, but the authors overlook the real-world applications to which QGST can be applied. KG exploration may be one such application, however, it isn’t a convincing one for me [D1].

W2. The theoretical complexity is still high, but it empirically performs efficiently. This should be discussed and explained in more detail. Besides, the approximation ratio has a similar issue, it’s too large to be meaningful theoretically, but it empirically is quite small, showing that this approximation ratio is a quite loose upper bound without real meaning [D2].

W3. A case study or user study would be necessary to evaluate how HB satisfy users’ demand on exploring a KG. Or, my concern is that the entire experiment is designed to show the speedup of HB, but ignore its effectiveness (i.e., lacking of accuracy evaluation) in real-world application [D3].

Detailed comments

D1. KG exploration may be one application of QGST, but QGST may not be the best solution for KG exploration. One can come up with a quite efficient graph query to return subgraphs that match with the different combination of input keywords. Or, we can just return the induced subgraphs of a set of keywords (in which each keyword comes from different group) and ranking them by cost(). A more straightforward and motivating application that is suited for QGST would be better choice than KG exploration. Moreover, the running example in Figure 2 is meaningless, it’s better to use a real example from existing KGs. Especially, the weights and q weights should be clarified with more real-meanings. Back to Figure 2, if a user wants to explore the subgraphs that related to or contain parts of input keywords, I would say the input graph (a) is a better result than two QGSTs in (c).

D2. The approximation ratio (g-1)^2\cdot n is quite a large value that one may think is meaningless because it indicates that the returned results have a large difference in cost() compared with the exact cost(). Note that n is the size of the entire graph (see Sec-2.1, 1st paragraph), making the approximation ratio unacceptable. Even if n is the number of keywords, (g-1)^2\cdot n is still too large to be a useful approximation ratio. However, Table 3 shows that the ratio is quite small, this proves that the proposed ratio is useless. Moreover, recalling the complexity, what’s the root reason why HB can respond within 1sec given a large time complexity, should be clarified.

D3. I am not sure how HB can support KG exploration in practice. Therefore, I expect a case study to demonstrate some interesting examples. Besides, I expect a user study to evaluate how the users are satisfied with the explored results, e.g., using PCC as a metric.

**Questions:**

Q1. Motivation should be clarified. What kind of applications can QGST be used with?

Q2. Why do we need such a bad approximation ratio for an approximation algorithm?

Q3. Any useful case study and user study can be provided?

**Reviewer Confidence:**

3: The reviewer is confident but not certain that the evaluation is correct

**Scope:**

4: The work is relevant to the Web and to the track, and is of broad interest to the community

---

### Official Review · Reviewer_qU5E · 2023-11-21

**Novelty:** 5
**Technical Quality:** 6

**Review:**

The paper considers an extension of the group Steiner tree problem: in an unweighted graph a root vertex $r$ is to be connected with $g$ groups of vertices, each vertex has a weight and there is a (pseudometric) function assigning weight to pairs of points. The objective function is then the sum of the vertex weights plus the sum of the pairwise weights, with the possibility of balancing the relative importance of the two contributions.
The paper proposes an algorithm with approximation factor $(g-1)^2 n$, the same as the state of the art, but with a much lower computational complexity.
The theoretical analysis is complemented by an experimental evaluation on several knowledge graphs.

The presentation of the algorithm is very notation-heavy, and in my opinion could greatly benefit from a more high level description that provides more intuition. Clearly the core idea is to build a Quadratic Group Steiner Tree by first finding a Relevant Path Set, and what the paper provides is an improved way to compute such a set, compared to the state of the art. However there is little intuition provided on how the paper does so. Section 3.1 (main idea) jumps straight to the definition of `hcost`, without giving an overview of how the algorithm will find small-hops paths, and why they are important.

The experimental section is well organized, with some parts that could still provide more insight. For instance, the results paragraph at the end of page 6 just repeats what can already be read in Table 2. It would be more interesting to use that space to answer the following question: why does the running time change when changing $\alpha$? In the time complexity $\alpha$ does not appear, therefore it would be interesting to discuss this phenomenon.

In the setup of the experimental what is the rationale for picking the angular distance between the embeddings? Most likely this is to reward trees using similar vertices, but this should be stated explicitly. Furthermore, why the embedding has 10 dimensions? For the purpose of this experimental section any number is of course fine, but what is the rationale for this choice?

**Questions:**

What is the main intuition underlying the algorithm, in four sentences?

**Ethics Review Description:**

-

**Reviewer Confidence:**

3: The reviewer is confident but not certain that the evaluation is correct

**Scope:**

4: The work is relevant to the Web and to the track, and is of broad interest to the community

---

### Official Review · Reviewer_QCou · 2023-11-24

**Novelty:** 4
**Technical Quality:** 5

**Review:**

The paper studies the problem of Quadratic Group Steiner Tree that identifies a semantically cohesive subgraph connecting entities. As an incremental work, this paper aims to design a fast approximate method. Extensive experiments on 10 graphs have been conducted.

Strong points:

S1: The paper proposes a faster approximation method for QGSTP.

S2: The overall writing is quite clear and fluent.

S3: Extensive experiments have been conducted.

Weak points:

W1: The motivation of QGSTP is not clear enough.

W2: The experiments can be enhanced.

Detailed comments:

D1. The motivation of QGSTP is not clear enough. In the example (Figure 1) the vertex-pair semantic distance is not clearly described. Precisely, the entities in T_1 are related to a common notion, which can be described as weights of vertices as well. Moreover, it is not known whether QGSTP can always beat GSTP. If not, the users may be confused about when to choose QGSTP.

D2. The paper stated that ``answer a query over a million-scale graph in about one minute’’, however, it may be arbitrary as the elapsed time also depends on the complexity of queries.

D3. It is highly recommended to include B^3F and QO for comparison.

D4. The queries used in the experiments may not be reasonable. For example, the queries for MND are not defined for the QGSTP.

D5. As the proposed method is approximate, it would be interesting to explore the effect of the cost of the QGST. Does small cost really lead to better QGST?

**Questions:**

see the comments above

**Reviewer Confidence:**

3: The reviewer is confident but not certain that the evaluation is correct

**Scope:**

3: The work is somewhat relevant to the Web and to the track, and is of narrow interest to a sub-community

---

### Official Review · Reviewer_iZh1 · 2023-11-24

**Novelty:** 5
**Technical Quality:** 6

**Review:**

This paper tackles the problem of Quadratic Group Steiner Tree (QGSTP).
The main contribution of the paper is a method which is is consistently faster than the fastest state-of-the-art method for QGSTP, while still being comparable/better in terms of effectiveness (both practical and theoretical).

The paper comes with several important strengths:

S1) The paper provides an authoritative advancement to the state of the art in QGSTP.

S2) The proposed method is well-designed and sound.

S3) The empirical efficiency and effectiveness of the proposed method are convincingly attested via a well-designed and comprehensive experimental evaluation.

In terms of weaknesses:

W1) There are a number of aspects in the paper that are not entirely clear, or require further investigation/actions. However, these are not really major deficiencies. Specifically:

 W1.a) It would nice to see ablation study results on more datasets (currently, the results on only one (small) datasets are reported (Table 4)).

 W1.b) It would be good to make it clear that Section 2.2 (including Lemma 1) is state of the art (it is taken -- almost as is -- from [25]).

 W1.c) Section 2.1: it is not specified whether the query sets K_1, ..., K_g are disjoint or may overlap.

 W1.d) Section 2.2, GenAns: what is it exactly meant by "structurally minimal sub-tree"? Just any spanning tree of G' or what?

**Questions:**

Please answer to W1.c) and W1.d).

**Ethics Review Description:**

No ethical issues.

**Reviewer Confidence:**

3: The reviewer is confident but not certain that the evaluation is correct

**Scope:**

4: The work is relevant to the Web and to the track, and is of broad interest to the community

---

### Decision · Program_Chairs · 2024-01-22

**Decision:**

Accept (Oral)

**Comment:**

The reviewers appreciated the clear goal of the paper - providing a new algorithm for the quadratic group Steiner tree problem that empirically outperforms the state-of-the-art solution. The algorithm, analysis, and experimental evaluation are structured and presented well in the paper. However, multiple reviewers raised doubts about the practical relevance of this abstract problem. Moreover, the theoretical analysis results in unimpressive bounds both in terms of approximation ratio and time complexity of the algorithm. While empirical evidence points to better performance in practice, the weak theoretical bounds add little value to the paper. Overall, the reviewers were of the opinion that in spite of the shortcomings, the paper would make a good addition to the conference. As a result, the paper is being weakly recommended for acceptance.